# Degenerate conic anchoring and colloidal elastic dipole-hexadecapole transformations

Ye Zhou [1], Bohdan Senyuk[2], Rui Zhang [1], Ivan I. Smalyukh [2,3,4] & Juan J. de Pablo[1,5]

The defect structure associated with a colloid in a nematic liquid crystal is dictated by molecular orientation at the colloid surface. Perpendicular or parallel orientations to the surface lead to dipole-like or quadrupole-like defect structures. However, the so-called elastic hexadecapole discovered recently, has been assumed to result from a conic anchoring condition. In order to understand it at a fundamental level, a model for this anchoring is introduced here in the context of a Landau-de Gennes free energy functional. We investigate the evolution of defect configurations, as well as colloidal interactions, by tuning the preferred tilt angle ($\theta_e$). The model predicts an elastic dipole whose stability decreases as $\theta_e$ increases, along with a dipole-hexadecapole transformation, which are confirmed by our experimental observations. Taken together, our results suggest that previously unanticipated avenues may exist for design of self-assembled structures via control of tilt angle.

[1] Institute for Molecular Engineering, The University of Chicago, Chicago, IL 60637, USA. [2] Department of Physics and Soft Materials Research Center, University of Colorado, Boulder, CO 80309, USA. [3] Department of Electrical, Computer, and Energy Engineering, Materials Science and Engineering Program, University of Colorado, Boulder, CO 80309, USA. [4] Renewable and Sustainable Energy Institute, National Renewable Energy Laboratory and University of Colorado, Boulder, CO 80309, USA. [5] Argonne National Laboratory, Argonne, IL 60439, USA. Correspondence and requests for materials should be addressed to Y.Z. (email: yezhou@uchicago.edu) or to I.I.S. (email: Ivan.Smalyukh@colorado.edu) or to J.P. (email: depablo@uchicago.edu)

Nematic colloids—colloidal particles immersed in a liquid crystal (LC) host—have been studied extensively over the past decade[1–10]. Seminal experiments by Poulin et al.[1] reported striking observations pertaining to strong and long-range structural forces between nematic colloids, which arise from the anisotropy of nematic media[11–13]. A more quantitative characterization of these elasticity-mediated forces was later conducted using a dual-beam laser trap[14,15]. With a deeper understanding of the structural forces that arise between colloids in nematic materials, it has now been possible to study the formation and characterization of a variety of self-assembled structures in one-dimension (linear chains)[1,2], two dimensions[3,6], and three dimensions[9,16], paving the way for potential applications in photonics[17]. Such assemblies have also had an impact in other scientific domains, including optical manipulation of nematic colloids[18,19], knot theory[20], and memory effects[21].

For years, only three possible defect configurations were assumed to arise in spherical nematic colloids. One with two surface point defects (boojums) at the poles, for colloids with tangential (degenerate planar) anchoring, Saturn-ring configurations with a disclination ring in the bulk at the equator, and dipolar configurations with a bulk point defect (hyperbolic hedgehog) for colloids having homeotropic anchoring[6,13,22–24]. Quadrupolar (boojums and Saturn-ring configurations) and dipolar (a hedgehog configuration) symmetries severely limit the possibilities for formation of colloidal bonds and self-assembly. Recently, an elastic hexadecapole was created by relying on an insightful analogy to electrostatic charge distribution[25], and it was suggested that the new symmetry stems from a degenerate conic anchoring condition imposed at the colloid surface (where conic alignment has easy axes along a conic surface at a specific polar angle ($\theta_e$)[26–28]).

To better understand this new defect structure, and to help design novel colloidal lattices, new models must be developed that are capable of describing this type of anchoring and its consequences for defect formation. With that goal in mind, a continuum model is introduced here for the order parameter tensor $\mathbf{Q}$ at a liquid crystal interface. By incorporating that model into a Landau-de Gennes theory, we are able to conduct a systematic study of nematic colloids with degenerate conic anchoring and the corresponding defect structures as a function of $\theta_e$. In the first section, we investigate the elastic hexadecapole at a quantitative level, and characterize its region of stability in terms of $\theta_e$. The angular and radial dependence of the elasticity-mediated interactions between colloids are unique for different $\theta_e$. Moreover, we go beyond the elastic hexadecapole and discover a new elastic dipole species for nematic colloids with degenerate conic anchoring, which is confirmed in our experiments. Our calculations provide useful insights concerning the meta-stability of this elastic dipole with increasing $\theta_e$, and its transition into an elastic hexadecapole, as observed in experiments.

## Results

**Elastic quadruple and hexadecapole.** Using the **Q**-tensor-based surface energy term (Eq. (5)), one can examine an individual particle ($R = 250$ nm) with degenerate conic anchoring confined in a nematic channel ($h = 1.5$ μm). The inclusion of colloids with curvature inevitably leads to orientational frustration, generating point defects, or disclination lines as a result. In agreement with previous experimental work[25], the nematic colloidal particle with degenerate conic anchoring ($\theta_e = 45°$) exhibits two boojums at the poles and a defect ring at the equator, resembling a combination of defect configurations of colloids with $\theta_e = 0°$ and 90° (Fig. 1a). Moreover, as shown in Fig. 1c, the $n_x$ color maps for nematic colloids with $\theta_e = 0°$ and 90° both exhibit quadruple symmetry, while that for $\theta_e = 45°$ shows an hexadecapolar symmetry, similar to a superposition of two quadruples of opposite sign.

In terms of polarized light micrographs, a uniform LC channel appears all dark when the far-field director $\mathbf{n}_0$ is parallel to the polarizer or analyzer, while the perturbation induced by the inclusion of colloids may lead to brightness. For instance, the lobes in the micrographs (Fig. 1b) for colloids with $\theta_e = 0°$ (homeotropic) and $\theta_e = 90°$ (degenerate planar) correspond to elastic distortions surrounding the disclinations. Interestingly, the polarization graph of a colloid with $\theta_e = 45°$ displays eight separated lobes near the colloid surface; these observations are consistent with the key optical features of the elastic hexadecapole reported in the recent literature[25]. More importantly, Fig. 1d shows that the relative brightness of the eight lobes varies with $\theta_e$, providing a potential methodology to measure $\theta_e$ in experiments (Supplementary Fig. 1). Note that, in Fig. 1b, the brightness for colloids with $\theta_e = 0°$ and 90° has been reduced by half for the purpose of comparisons, implying greater elastic deformations compared to those in elastic hexadecapoles. This is further illustrated by the elastic free energy analysis of nematic colloids with increasing $\theta_e$, which shows a minimum of elastic energy near $\theta_e = 60°$ (Fig. 1e).

At the next level of complexity, we consider the elasticity-mediated interactions that arise between nematic colloids with different $\theta_e$, which are central to understanding colloidal self-assembly. The two particles are confined by a uniform LC channel. As illustrated in Fig. 2b, the inter-particle separation is denoted by $d$ and the angle between the uniform far-field nematic director and particle–particle vector is given by $\alpha$.

As a means to validate our calculations, we first consider the elastic interactions between colloids with degenerate planar anchoring ($\theta_e = 90°$). Figure 2a shows the free energy for a two-particle system, at different colloidal separations $d/R$ and different orientations $\alpha$. As colloidal separation $d/R$ increases from 2.4 to 3.4, the $\alpha$ corresponding to the minimum energy, defined as $\alpha^*$, gradually shifts from 30° to 45°, as shown in the inset of Fig. 2a. The vector field of forces in Fig. 2b shows more clearly that the colloids attract each other for $\alpha < 70°$ and repel each other for $\alpha > 70°$ with $d/R = 2.6$. As particles move apart from each other, the forces become weaker, and the attraction direction migrates to $\alpha$ near 45° for $d/R = 3.2$. These findings are in agreement with past literature reports[15,29].

In order to understand how elasticity-mediated interactions change as the preferred tilt angle $\theta_e$ is varied, we fix the colloidal separation ($d/R = 2.4$) and plot the angular dependence for interactions as a function of $\alpha$, for different $\theta_e$. Results are shown in Fig. 2c. Similar to the colloids with degenerate planar anchoring, colloids with $\theta_e = 0°$ have one energy well at $\alpha^* = 65°$, consistent with past reports[6]. Instead of a simple shift of the energy well from $\alpha^* = 65°$ ($\theta_e = 0°$) to $\alpha^* = 30°$ ($\theta_e = 90°$), a double-well state appears between $\theta_e = 40°$ and 60° (Fig. 2c). It is within the same range of $\theta_e$, where the eight-lobe pattern in polarized optical images (Fig. 2d) becomes pronounced. At $\theta_e$ ~55°, the depths of the two energy wells are approximately identical; as shown in the inset of Fig. 2c, the $\alpha^*$ corresponding to the deepest well switches from 70° to 20° at $\theta_e$ ~55°.

The dependence of this double-well energy profile on colloidal separation $d$ is examined in closer detail for colloids with $\theta_e = 45°$. Results are shown in Fig. 2d, e. In contrast to colloids with simple degenerate planar anchoring, the two energy wells corresponding to the elastic hexadecapoles are basically localized near $\alpha = 20$–25° and $\alpha = 70$–75° for increasing colloidal separations. This indicates that the forces between elastic hexadecapoles, be they attractive or repulsive, are basically along the radial direction. Taken together, our results therefore show that both the nature of colloidal interactions (attractive/repulsive) and their

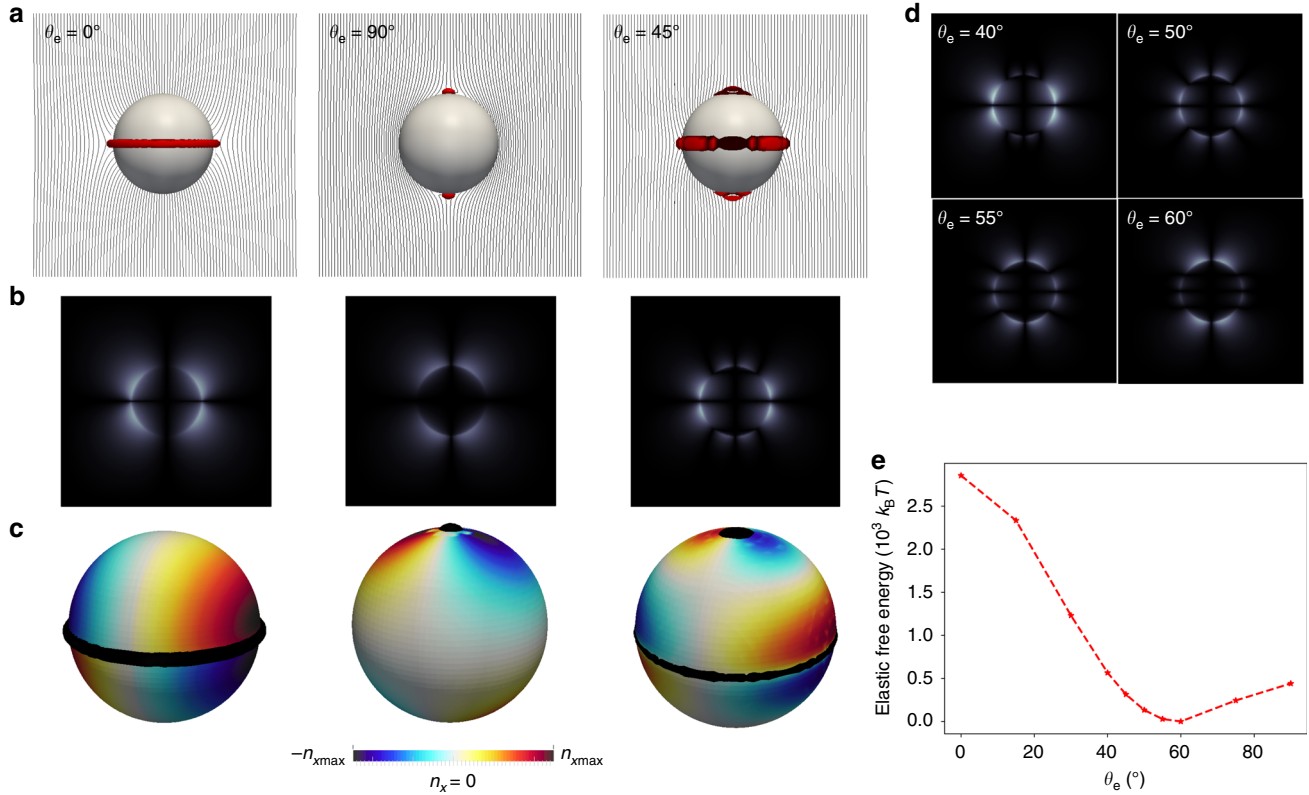

**Fig. 1** Elastic quadruple and hexadecapole. **a** Director fields for nematic colloids ($R = 250$ nm) with $\theta_e = 0°$, $90°$, and $45°$, respectively. Defects are shown in red (isosurface for $S = 0.6$). **b** Corresponding simulated polarized light micrographs for nematic colloids with $\theta_e = 0°$, $90°$, and $45°$. The brightness of the images for $\theta_e = 0°$ and $90°$ is reduced by half for the purpose of comparisons. The far-field director is parallel to the polarizer or analyzer. **c** Color map of the directors' $x$-component ($n_x$) on the colloid surface. **d** Simulated polarized light micrographs for nematic colloids with $\theta_e = 40°$, $50°$, $55°$, and $60°$. **e** Elastic free energy for nematic colloids with increasing $\theta_e$

dependence on $\alpha$ and $d$ are governed by the angle $\theta_e$ at the colloid surface. This finding suggests that new avenues that rely on manipulation of this angle may be used to control the formation of new and diverse colloidal assemblies.

A similar analysis of the elastic multipole moments as that performed in recent experiments[25] is conducted here by a least-squares fit to the theoretical colloidal pair-interactions (Eq. (1)), derived from an electrostatic analogy of the far-field director distortions. Here the energy is given by,

$$U_{int} = 4\pi K \sum_{l,l'=2,4,6} a_l a_{l'}' (-1)^{l'} \frac{(l+l')!}{d^{l+l'+1}} P_{l+l'}(\cos\theta) \quad (1)$$

where $a_l = b_l R^{l+1}$ represents the elastic multipole moment of the $l$th order ($2^l$-pole), and $K$ is an average Frank elastic constant.

The ratios of elastic quadruple moment ($b_2$) to hexadecapole moment ($b_4$) obtained from fitting (Fig. 2f) are 0.61, 5.57, and $-9.53$ for nematic colloids with $\theta_e = 45°$, $90°$, and $0°$, respectively. These results serve to emphasize the fact that quadruple moments (with opposite signs) are dominant at $\theta_e = 0°$ and $90°$, and they cancel each other at around $\theta_e = 45°$, thereby letting the hexadecapole symmetry stand out.

### Elastic dipole and its transition to hexadecapole

By initializing a different, specific condition (Eq. (11)), we also predict another candidate structure for nematic colloids having degenerate conic anchoring (CA): we refer to this structure as an 'elastic CA dipole' to distinguish it from the more commonly studied elastic dipole formed by colloids with perpendicular surface boundary conditions.

As shown in Fig. 3a, a nematic colloid with homeotropic anchoring ($\theta_e = 0°$), which adopts a homeotropic anchoring (HA) dipole configuration, exhibits a bulk hedgehog defect at the upper pole. As the anchoring becomes conic ($\theta_e > 0°$), a boojum emerges at the lower pole (e.g., Fig. 3b). Since all the structures of the director field around colloidal particles considered here have axial symmetry with respect to the far-field director, for simplicity, it is possible to analyze topological charge conservation in terms of 2D defect topological charges within the plane containing $\mathbf{n}_0$. In terms of such 2D charges, it is known that colloids with homeotropic anchoring carry a charge of $q = +1$; since the charge of the hedgehog defect in a dipolar configuration is $q = -1$, the total equals zero[4]. Nematic colloids with degenerate conic anchoring also carry a 2D effective charge of $q = +1$ (Supplementary Fig. 2). Figure 3b shows the defect configuration for a nematic colloid with $\theta_e = 45°$, where the defect strength ($-1$) is unevenly distributed amongst upper ($-3/4$) and lower surface defects ($-1/4$) while complying with the conservation of charges, as before. The formation of a boojum in nematic colloids with degenerate conic anchoring can also be visualized in simulated polarized light micrographs (Fig. 3c, d). The position of the two lobes on top of the colloids exhibits another significant distinguishing feature: for HA dipoles (Fig. 3c), the lobes spread out evenly along the vertical axis, while those in CA dipoles (Fig. 3d) are closer to the upper defect of $-3/4$.

Experiments on colloidal particles were used to verify the predictions outlined above (Fig. 3b, d). Most of the observed droplets had homeotropic anchoring with a typical dipolar configuration of the director (Fig. 3a, c, e). However, a small number of particles showed a hexadecapolar texture (Fig. 1), characteristic

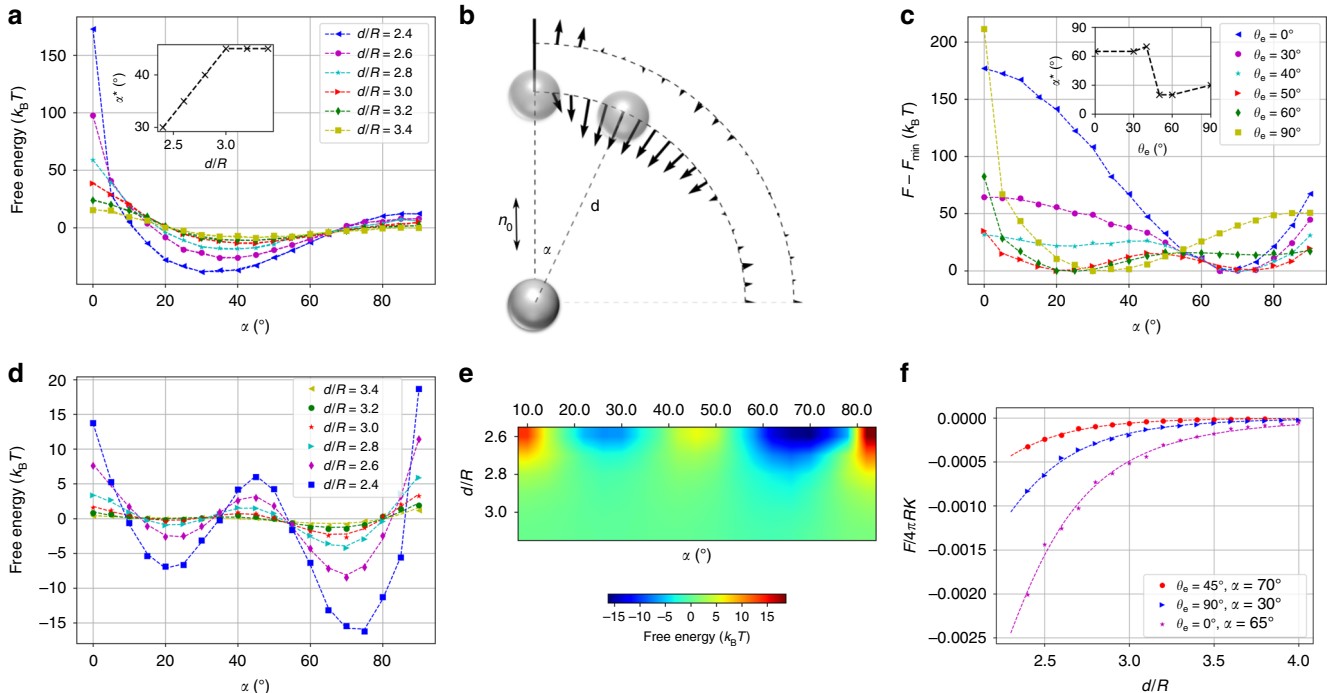

**Fig. 2** Angular and radial dependence of colloidal interactions. **a** Free energy as a function of $\alpha$ for different inter-particle separations $d/R$ for $\theta_e = 90°$ for two particles ($R = 200$ nm) confined in a uniform LC channel ($h = 3.6$ μm). The inset plot shows the dependence of $\alpha^*$ on the colloidal separation $d$. **b** Vector field of forces between two colloidal particles for $d/R = 2.6$ and 3.2, derived from the spatial gradient of free energy. The far-field director is denoted by $\mathbf{n}_0$. **c** Evolution of free energy as a function of $\alpha$ with different $\theta_e$ ($d/R = 2.4$). The inset plot shows the dependence of $\alpha^*$ on colloidal separation $d$. **d**, **e** Evolution of free energy as a function of $\alpha$ for different $d/R$ with $\theta_e = 45°$. **f** Dependence of attraction on $d/R$ for particles with $\theta_e = 45°$, 90°, and $\theta_e = 0°$, along $\alpha = 70°$, 30°, and 65°. The dashed line is a least-squares fit of colloidal pair-interactions. The coefficients are ($b_2 = -0.0088$, $b_4 = -0.0144$, $b_6 = -0.0004$), ($b_2 = -0.0496$, $b_4 = -0.0089$, $b_6 = -0.0005$), ($b_2 = 0.1001$, $b_4 = -0.0105$, $b_6 = 0.0009$) for colloids with $\theta_e = 45°$, 90°, and $\theta_e = 0°$, respectively

of conic anchoring, or a texture with a hedgehog point defect at one pole of the droplet and a boojum point defect at the other pole (Fig. 3b, d, f, g). At first sight, the latter director field around the droplet resembles the dipolar configuration around particles with homeotropic anchoring. However, upon more careful examination, one can see that the elastic CA dipole observed in this experiment has several subtle differences. First, there are two weakly bright lobes surrounding the dark point of the boojum defect at the pole opposite to the one with the hedgehog (compare Fig. 3e–g; pointed by Arrow 1). The dark point corresponding to the boojum defect is poorly visible in the textures with parallel polarizers, as the scattering from it is weak and blends together with strong scattering from the droplet contour. Secondly, there is a sequence of dark and bright brushes at the pole with a hedgehog in the typical HA dipole (pointed out by Arrow 2). Lastly, two bright areas within the polarized texture of the droplet and two corresponding dark areas in the texture between parallel polarizers (pointed respectively by arrows 3 and 4 in Fig. 3e–g) extend from the top to the bottom pole in the typical HA dipole. However, in the elastic CA dipole in our experiments, they are located mostly in the hemisphere with the hedgehog defect. The good agreement between polarized light textures of the elastic CA dipole observed in the experiments and predicted in our calculations serves to confirm the conic anchoring that we have at the surface of the droplets and, importantly, supports the proposed model of surface free energy for degenerate conic anchoring.

It is difficult in experiments to observe the elastic CA dipole for large $\theta_e$. Upon disturbing the system, it evolves spontaneously into a more stable elastic hexagecapole (Fig. 4j). Our calculations also predict a vanishing energy barrier between elastic CA dipoles and hexagecapoles as $\theta_e$ increases and becomes larger than 45°.

This is to be expected, because as $\theta_e$ increases to 90°, the nematic colloid has no choice but to adopt a quadrupolar symmetry. To understand this transition, we perturb an equilibrium CA dipole configuration with $\theta_e = 45°$ by setting $\theta_e$ to 60°, and monitor the ensuing relaxation process (Fig. 4a–h). In the early stages, the CA dipolar colloid with $\theta_e = 60°$ distributes the 2D charge $q = -1/3$ to the lower pole defect and $q = -2/3$ to the upper pole defect (Fig. 4a). Subsequently, a surface defect ring of $q = -1/3$ splits out from the upper pole defect and gradually migrates downwards. As the defect ring arrives at the equator, it forms an elastic hexagecapole (Fig. 4d).

Figure 4i provides a closer look at the director field near the defect ring. Since the director profiles along the $x$-$z$ cross section are all in-plane, the easy cone that is observed in 3D collapses onto two easy axes (clockwise and counter-clockwise) for $\theta_e = 60°$. In the CA dipolar configuration, the directors lie along the clockwise easy axis on the right and along the counter-clockwise easy axis on the left side of the colloid surface, varying continuously from one pole to the other. Figure 4i shows that the directors on top of the defect ring (left side) have flipped from their original counter-clockwise to their clockwise easy axis. Therefore, the transition proceeds by gradually flipping the directors between two easy axes along the $z$-axis. As a consequence, we observe that the defect ring, which results from the orientational discontinuity at the flipping boundary, moves towards the equator.

The transition from the CA dipolar to the predicted hexagecapolar structure is also observed in the experiments (Fig. 4j). Right after filling the dispersion into the observation cells, the CA dipolar structure described above (Fig. 4a, e) prevails around droplets with conic anchoring. The boojums resulting from conic

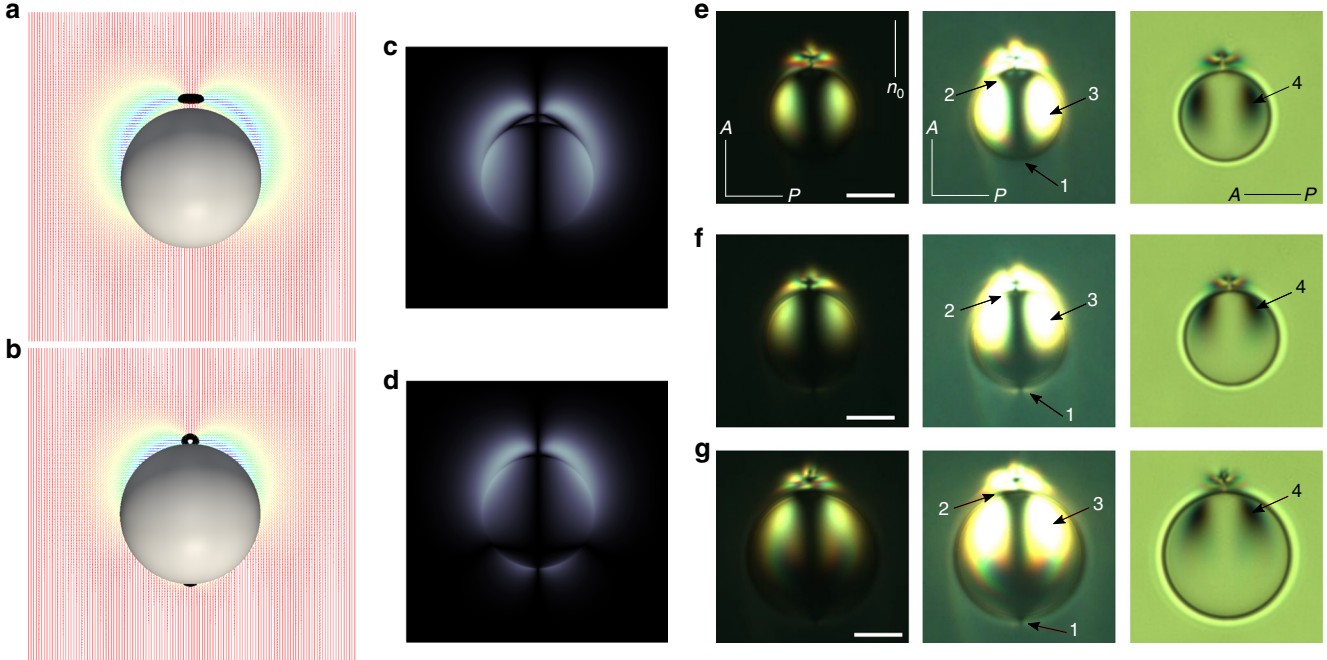

**Fig. 3** Elastic dipoles. **a**, **b** Director fields for dipole colloids ($R = 750$ nm) with $\theta_e = 0°$ (**a**) and 45° (**b**), colored by its projection onto the $z$-axis. The defects are shown in black (isosurface for $S = 0.6$). **c**, **d** Corresponding simulated polarized light micrographs of dipole colloids with $\theta_e = 0°$ (**c**) and 45° (**d**). **e–g** Optical microscopy textures of a dipole with homeotropic (**e**) and conic (**f**, **g**) anchoring, which are consistent with the predicted textures shown in (**a**, **c**) and (**b**, **d**), respectively; $\mathbf{n}_O$ shows a far-field director set by rubbing. Left and middle textures in **e–g** were taken between crossed polarizers A and P; textures in the middle are slightly overexposed to enhance visibility of boojums at the bottom pole. Textures in the right column were taken between parallel polarizers. Scale bar: 5 μm (**e**, **f**) and 6 μm (**g**)

anchoring scatter less light when compared to typical boojums in particles having tangential anchoring[15]; note that their weak scattering blends with strong scattering from the droplet's contour, which explains why boojums are only slightly visible in the bright field textures of our droplets. However, bright lobes around the boojums, caused by having the director field around them be tilted with respect to $\mathbf{n}_O$, are clearly visible in textures taken between crossed polarizers. After a while, that CA dipolar structure (see a texture at $t_0$) spontaneously changes to the hexadecapolar configuration (see a texture at $t_5$). The transition starts at the top pole from the hedgehog, breaking up into a boojum, similar to the one on the bottom pole, and a surface disclination ring (see a texture at $t_2$ and the calculated director field in Fig. 4i). A surface disclination ring, which can clearly be appreciated in the microscopic textures as a darker, blurred line, gradually extends and moves towards the equator of the droplet ($t_1 - t_5$). The transition is quite slow, and is completed when the surface disclination reaches the equator of the droplet ($t_5$), within 5–10 min from the start of the process. The predicted textures for the structural transition are in good agreement with experiment, serving to underscore the validity and the significance of the proposed model for the surface free energy under degenerate conic anchoring.

## Discussion

A model has been proposed here for degenerate conic anchoring at liquid crystal interfaces. With that model, it has been possible to investigate systematically two types of nematic colloids, with $0° \leq \theta_e \leq 90°$. For these anchoring angles, one forms an elastic quadrupole/hexadecapole and an elastic dipole. The elastic hexadecapole was first reported in recent experiments[25]; our calculations are consistent with such measurements. Equipped with this model, we were able to vary $\theta_e$ continuously, thereby going

beyond past experimental observations. New dipolar configurations have been identified, and these structures were confirmed in experiments, serving to highlight the usefulness of the proposed model. Note that a surface term, which represents an anchoring degenerate in the azimuthal angles we proposed, is crucial in this work. Especially, as the bend/twist anisotropy ($k_{33}/k_{22}$) increases, we show in simulations that the surface director can escape from its meridian plane and a chiral CA dipole with spontaneous twist is observed with $\theta_e = 45°$ (Supplementary Fig. 3). From a theoretical point of view, we have introduced an explanation for our observations that relies on vanishing energy-barriers for the meta-stability of certain structures with increasing $\theta_e$, and we have followed this transition to an elastic hexadecapole as $\theta_e >$ 45°. By monitoring the destabilization of the elastic conic anchoring dipole, we have also proposed a transition mechanism based on 'director-flipping-boundary defect rings'. That mechanism was also confirmed in our experiments. Beyond single colloid structures, we have showed that each $\theta_e$ defines a unique defect configuration, with a corresponding polarized light micrograph, and a specific angular/radial dependence for two-body colloidal interactions. Based on the relative brightness of the eight-lobe pattern in the polarized micrographs for colloids with different $\theta_e$, for example, one may engineer new approaches to measure the preferred tilt angle on the colloid surface via image-recognition techniques. Moreover, the improved understanding of specific inter-particle interaction for each $\theta_e$ presented here offers the potential to define new protocols for design of self-assembled nematic colloid structures having new symmetries.

## Methods

**Simulation details.** A Landau-de Gennes (LdG) continuum model is adopted here for the **Q** tensor, defined by $Q_{ij} = S\left(n_i n_j - \frac{1}{3}\delta_{ij}\right)$. Here $n_i$ are the $x, y, z$ components of the local director vector and $S$ is the scalar order parameter[30]. The bulk

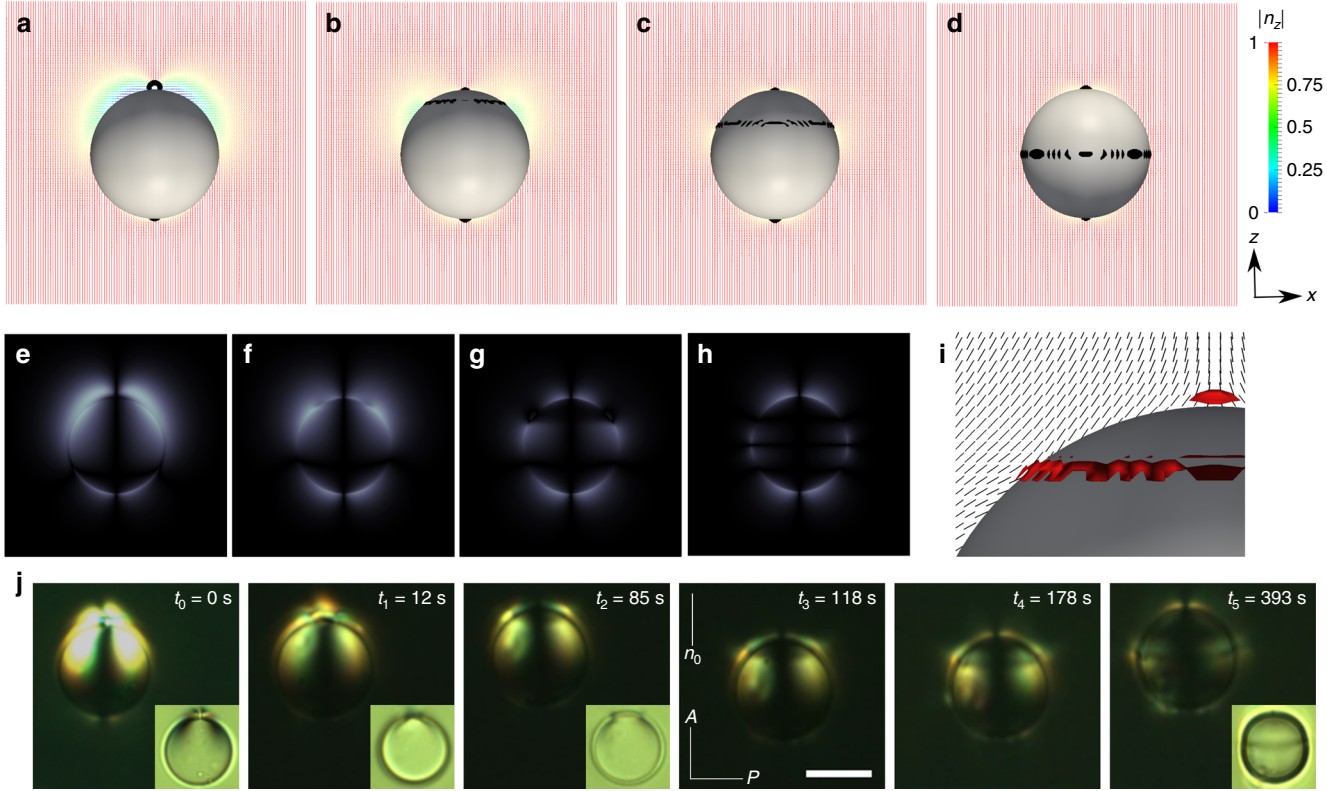

**Fig. 4** Dipole-hexadecapole transition. **a–d** A temporal sequence of configurations during relaxation after applying $\theta_e = 60°$ to an equilibrium elastic dipole of $\theta_e = 45°$. The director field is colored by its projection onto the z-axis, and the defects are shown in black (isosurface for S = 0.6). **e–h** Corresponding simulated polarized light textures. **i** Director field near the defect ring in **b**. The director field is shown in black and the defects are shown in red (isosurface for S = 0.6). **j** Experimental sequence of microscope textures showing the transition from a dipolar to a hexadecapolar structure taken between crossed polarizers. The first texture is slightly overexposed compared to the others in order to enhance the visibility of a boojum defect at the bottom pole. Insets show corresponding textures taken between parallel polarizers. The size of the bottom side of the inset image is 9.5 μm. Scale bar: 5 μm

free energy is given by:

$$f_{\text{bulk}} = \int_{\text{bulk}} \left( \frac{A}{2}\left(1 - \frac{U}{3}\right)Q_{ij}Q_{ji} - \frac{AU}{3}Q_{ij}Q_{jk}Q_{ki} \right.$$
$$\left. + \frac{AU}{4}\left(Q_{ij}Q_{ji}\right)^2 \right) \text{dV}$$
$$+ \int_{\text{bulk}} \frac{L}{2}\frac{\partial Q_{ij}}{\partial x_k}\frac{\partial Q_{ij}}{\partial x_k} \text{dV}, \tag{2}$$

where $A$ and $U$ are material constants, and $L$ is the elastic constant under the one-constant approximation. The first term corresponds to the phase free energy, which controls the equilibrium value of the order parameter $S_{\text{eq}} = \frac{1}{4}\left(1 + 3\sqrt{1 - \frac{8}{3U}}\right)$. The second term represents the elastic free energy, which governs long-range distortions of the director[31,32].

The free energy functional for planar degenerate anchoring was introduced by Fournier and Galatola[33]:

$$f_{\text{surf}}^p = \int_{\text{surf}} W_p \left(\tilde{Q}_{ij} - \tilde{Q}_{ij}^\perp\right)^2 \text{d}\Sigma, \tag{3}$$

where $W_p$ is anchoring strength, which generally ranges from $10^{-7}$ to $10^{-3}$ J/m². The projection operator is denoted by $P_{ij} = \delta_{ij} - \nu_i\nu_j$ and $\nu$ is the surface normal. The term $\tilde{Q}_{ij}^\perp = P_{ik}\tilde{Q}_{kl}P_{lj}$ is the projection of $\tilde{Q}_{ij} = Q_{ij} + \frac{1}{3}S_{\text{eq}}\delta_{ij}$ onto the plane perpendicular to $\nu$. Accordingly, surface molecules favor a tangential alignment with no in-plane preference.

In the literature, uniform surface anchoring is usually modeled by a Rapini-Papoular-like surface free energy expression[34]:

$$f_{\text{surf}}^h = \int_{\text{surf}} \frac{W_h}{2}\left(Q_{ij} - Q_{ij}^0\right)^2 \text{d}\Sigma, \tag{4}$$

where the preferred order parameter tensor at the surface, corresponding to a surface director aligned with the easy axis, is denoted by $Q_{ij}^0$. Such a model is capable of describing mono-stable anchoring with arbitrary easy axes $\mathbf{n}_e$ by penalizing quadratically any deviations of the director $\mathbf{n}$ from $\mathbf{n}_e$. Therefore, for $\mathbf{n}_e \neq \nu$, the anchoring is tilted, but it is non-degenerate. When the easy axis is along the surface normal, Eq. (4) yields a homeotropic anchoring condition.

In this work, we propose a different surface energy term to represent degenerate conic anchoring, given by

$$f_{\text{surf}}^c = \int_{\text{surf}} W_c \left(P'_{ik}\tilde{Q}_{kl}P'_{lj} - S_{\text{eq}}\cos^2\theta_e P'_{ij}\right)^2 \text{d}\Sigma. \tag{5}$$

Here $W_c$ is the anchoring strength for degenerate conic anchoring, $P'_{ij} = \nu_i\nu_j$ is an operator tensor similar to the projection operator $P_{ij}$ in Eq. (3), and $\theta_e$ is the polar angle between the conic surface and surface normal $\nu$ (Fig. 5a). In order to gain a better understanding of the **Q**-tensor-based surface free energy (Eqs. (3)–(5)), we transform these equations into $\theta_s$-expressions as explained below (derivations are provided in the SI), assuming that there is no spatial variation of the scalar order parameter ($S \equiv S_{\text{eq}}$):

$$f_{\text{surf}}^{p,\theta_s} = \int_{\text{surf}} W_p S_{\text{eq}}^2 \left(1 - \sin^4\theta_s\right) \text{d}\Sigma; \tag{6}$$

$$f_{\text{surf}}^{h,\theta_s} = \int_{\text{surf}} W_h S_{\text{eq}}^2 \sin^2\theta_s \text{d}\Sigma; \tag{7}$$

$$f_{\text{surf}}^{c,\theta_s} = \int_{\text{surf}} W_c S_{\text{eq}}^2 \left(\cos^2\theta_s - \cos^2\theta_e\right)^2 \text{d}\Sigma. \tag{8}$$

Here $\theta_s$ is the angle between the surface director and surface normal, defined as $\cos\theta_s = n_i\nu_i$.

The transformation reveals the fact that the surface free energies, $f_{\text{surf}}^h$, $f_{\text{surf}}^p$, and $f_{\text{surf}}^c$, are minimized when $\theta_s = 0$, $\pi/2$, and $\theta_e$, consistent with their anchoring types, i.e., homeotropic, degenerate planar, and degenerate conic anchorings (Fig. 5a), respectively. Here we note that other, simpler $\theta$-based expressions for degenerate conic anchoring have been employed in the literature[26,27]. Such expressions do not allow one to describe the defects engendered by conic anchoring. The **Q**-tensor expression proposed here, however, is advantageous in that it can capture the surface ordering, thereby permitting a description of the defects that arise in our systems, which is essential for our work and, more generally, for detailed studies of nematic colloids. Figure 5c shows the evolution of $\left(P'_{ik}\tilde{Q}_{kl}P'_{lj} - S_{\text{eq}}\cos^2\theta_e P'_{ij}\right)^2$ in

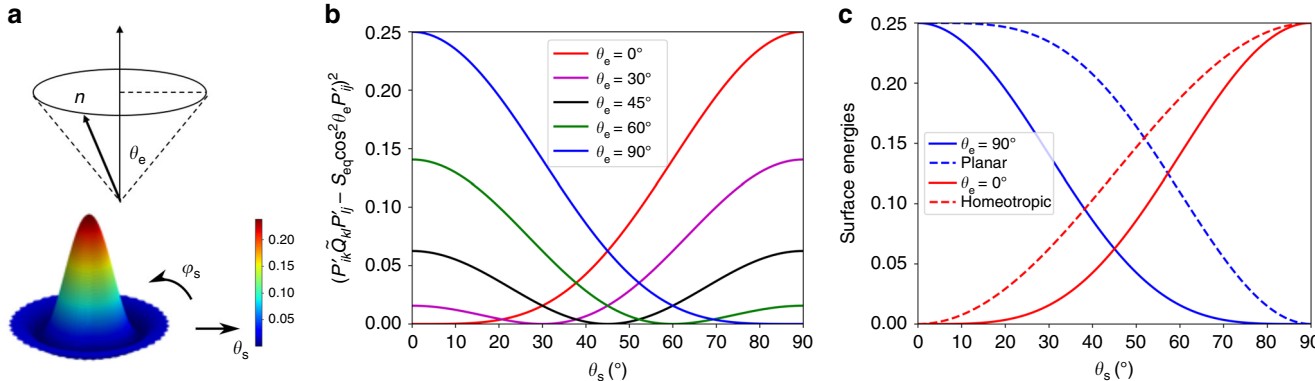

**Fig. 5** Degenerate Conic anchoring. **a** Schematic representation of degenerate conic anchoring with preferred tilt conic angle $\theta_e$. **b** The value of $\left(P'_{ik}\tilde{Q}_{kl}P'_{lj} - S_{eq}\cos^2\theta_e P'_{ij}\right)^2$ as a function $\theta_s$ and $\phi_s$ when $\theta_e = 60°$, which illustrates the degenerate conic anchoring with degeneracy in the azimuthal angle. **c** Values of $\left(P'_{ik}\tilde{Q}_{kl}P'_{lj} - S_{eq}\cos^2\theta_e P'_{ij}\right)^2$ as a function of $\theta_s$ for $\theta_e = 0°$, 30°, 45°, 60°, and 90° with $S_{eq} = 0.5$. **d** Comparison between the surface free energy term of $\frac{1}{2}\left(Q_{ij} - Q^0_{ij}\right)^2$ in Eq. (4), $\left(\tilde{Q}_{ij} - \tilde{Q}^\perp_{ij}\right)^2$ in Eq. (3), and $\left(P'_{ik}\tilde{Q}_{kl}P'_{lj} - S_{eq}\cos^2\theta_e P'_{ij}\right)^2$ in Eq. (5) with $\theta_e = 0°$, 90° as a function of surface tilt angle ($S_{eq} = 0.5$).

$f^c_{surf}$ as a function of surface director tilt angle $\theta_s$ for a range of preferred tilt angles. Again, these results demonstrate that, guided by the **Q**-tensor-based Eq. (5) proposed here, the surface directors favor orientation along a conic plane with an angle $\theta_e$ to the surface normal.

When compared to Eqs. (3) and (4), Eq. (5) provides a universal expression for surface free energy, since, in some sense, homeotropic and degenerate planar surface anchorings represent individual cases of degenerate conic anchoring for a 'collapsed cone' ($\theta_e = 0°$) or for a 'flat cone' ($\theta_e = 90°$):

$$f^{c,\theta_s}_{surf} = \begin{cases} \int_{surf} W_c S^2_{eq} \sin^4\theta_s \, d\sum . & \theta_e = 0°, \\ \int_{surf} W_c S^2_{eq} \cos^4\theta_s \, d\sum . & \theta_e = 90°, \end{cases} \quad (9)$$

Figure 5d illustrates the distinction between these expressions; degenerate conic anchoring (Eq. (5)) imposes a lesser energy penalty for deviations of the directors from their preferred direction than Eqs. (3) and (4).

An iterative Ginzburg-Landau relaxation (Eq. (10)) with finite differences on a cubic mesh of 7.15 nm is applied here to minimize the total free energy[35] (see detailed equations in SI):

$$\Gamma\frac{dQ_{ij}}{dt} = h_{ij}, \quad (10)$$

where $\Gamma$ is a numerical relaxation constant and $h_{ij}$ is referred to as a molecular field.

In order to ensure the formation of a dipolar nematic colloid, and to speed up the relaxation process, a specific initial condition is introduced according to:

$$\mathbf{n} = \mathbf{n}_0 + PR^2 \frac{\mathbf{r} - \mathbf{r}_{col}}{|\mathbf{r} - \mathbf{r}_{col}|^3}, \quad (11)$$

where $\mathbf{n}_0$ is the unit vector for the far-field director; $P = 2.1$ is a constant, determining the initial position of the point defect; $R$ is the colloidal radius; $\mathbf{r}$ is the position vector for the current position, and $\mathbf{r}_{col}$ is the position vector for the colloid center. The plus sign between the two terms defines the direction of the dipole. The tensorial order parameter **Q** is initialized by $S_{eq}$ and **n** as defined above.

Polarized light micrographs were calculated using the Jones matrix formalism, in which light traverses along a chosen direction and the total phase shift is accumulated[36]. In all cases, the polarizer and analyzer are placed perpendicular to each other, with the polarizer parallel to the z-axis. The light wavelength used in this work is 351 nm.

The following numerical parameters were used: $A = 1.17 \times 10^5 \,\text{J m}^{-3}$, $U = 3.5$, $L = 6 \times 10^{-12} \,\text{N}$, $W_c = 10^{-3} \,\text{J m}^{-2}$, unless specified otherwise. The channels in this work have periodic boundaries along the x and y axes, and rigid homeotropic anchoring along the z-axis.

**Experimental details.** Glycerol (Sigma–Aldrich) droplets were suspended in a nematic LC 5CB (4-cyano-4′-pentylbiphenyl from Frinton Laboratories, Inc.) with a small amount (<0.1 vol.%) of a molecular surfactant (sodium dodecyl sulfate) mixed with glycerol. The mixture of glycerol (about 10 vol.%) and 5CB was vigorously stirred to obtain glycerol spherical droplets ($R \approx 1$–10 μm) dispersed evenly in the LC host. Dispersions were filled into ~30-μm thick cells made of two glass plates separated by glass spacers and sealed with a UV-curable glue. A polyimide PI2555 (HD

Microsystems) was spin-coated on the glass plates, baked at 270 °C and uni-directionally rubbed with a velvet cloth to create a homogeneous planar alignment of the LC. Experimental samples were stable over at least several weeks and the director structures around the droplets were studied using bright field and polarized light optical observations with a ×100(NA = 1.42) oil objective mounted on an inverted Olympus IX81 microscope. Image acquisition and analysis were performed using a CCD camera (Flea, PointGrey) and ImageJ software, respectively.

## Data availability
Data and analysis codes are available from the authors upon request.

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

## Acknowledgements

Y.Z. and J.J.d.P. acknowledge support from the Department of Energy, Office of Basic Energy Sciences, Division of Materials Sciences and Engineering, Biomaterials Program under Grant No. DE-SC0004025, for development of models of liquids crystals and their mixtures that incorporate detailed anchoring conditions. Additional support for development of public domain software for simulation of liquid crystals from the Department of Energy, Basic Energy Sciences, Materials Science and Engineering, through the Midwest Integrated Center for Computational Materials (MICCoM) is also gratefully acknowledged. B.S. and I.I.S. acknowledge partial support from the NSF under grant DMR-1420736.

## Author contributions

Y.Z., B.S., I.I.S., and J.J.d.P. designed the research; Y.Z. and J.J.d.P. developed the models and performed the simulations; B.S. and I.I.S. conducted the experiments; R.Z. contributed to the theoretical analysis; Y.Z., B.S., R.Z., I.I.S., and J.J.d.P. wrote the manuscript.

## Additional information

**Competing interests:** The authors declare no competing interests.

