## [Peer Review File · Nature Communications]

Reviewers' comments:

Reviewer #1 (Remarks to the Author):

The Authors present an extension of their finding in ref 25. They apply Landau-de Gennes model to describe the director and scalar order at and around a sphere imposing a conically degenerate easy axis at its surface.

Whereas ref 25 presented a discovery of a dodecapole particle as well as the main physical mechanisms and conditions behind its existence, the present paper cannot pretend for a similar level of impact and novelty. I therefore consider it as not suitable for Nature.

Of course, this does not undermine the fact that the subject of the paper is very interesting. It presents a great piece of experimental work which has been performed at a very high level. However, I have found that the theoretical part and the general idea of conic degenerate anchoring as the necessary condition for the effect to occur should be revised.

First, writing down an anchoring term does not account to a new model, the idea behind it is simple. In particular, calling P' a new projection tensor does not sound quite appropriate. This is a minor point.

The most serious problem is in the form of anchoring term (4). It is known that when all other forces are absent, the anchoring term is minimized by $Tet_s = Tet_e$, hence its expansion in $(Tet_s - Tet_e)$ must begin with a quadratic term. In contrast, the eq 7 produces a linear term $\sim \sin(Tet_s - Tet_e)$ which is incorrect. The correct term, I believe, is $\sim \sin^2(Tet_s - Tet_e)$.

Further, if S can change then it can depend on the distance from the surface as well. Why strong deviations of n from n_e at the defect cores does not induce $gradS$ along the surface normal to decrease the anchoring? To incorporate such $gradS$ the anchoring term has to be modified into a bulk one.

Finally, I see no reason for the special need of an azimuthally degenerate anchoring. This kind of anchoring is needed when the director turns about the surface normal to lower the energy cost. In the case of our interest this would mean to lower the cost of a f_i director component. But in all of the figures both in the present manuscript and in ref 25, I see the director to lie solely in the meridian planes without any f_i component. Therefore, I conclude that to explain the effects presented it is sufficient to have a tilted easy axis lying in the meridional planes. Moreover, the anchoring can be just planar but not very strong that the cost of director's leaving the surface be compatible with the gain due to smoother distortions at the defect cores. In addition, I have not noticed specific means to get a conic anchoring at otherwise standard spherical particles. Is this so trivial? My knowledge has been that producing this kind of anchoring is a highly nontrivial problem.

In this connection I may also note that an azimuthal f_i director component (which would justify the degeneracy) on a sphere could give rise to a chiral dipolar component as described in Phys.Rev.E 84, 031702 (2011), and this would be experimentally detectable.

I also note that the fact that only three configurations around spherical particles with normal and planar anchoring are possible implies that the anchoring strength is infinite. If it is finite, then the configuration can be defectless. Or, maybe, of a hexadecapole type.

The Authors have to address the above problems priory to any publication venue.

Referee Report

The authors of the manuscript develop a new elastic surface energy to model degenerate conic anchoring. Later, this expression is used to analyze the static liquid crystal director field around a single colloidal particle, and after it, in a system composed of two of them. Finally, they study the transition between a state with dipolar defects to one with hexadecapolar one.

In general the article is well written. The abstract clearly shows the subject of the article and the introduction poses the problem very comprehensibly. The manuscript shows new results which are relevant to both basic and applied liquid crystal research. Moreover, all theoretical predictions are supported by experimental results. However, the section “Model and Methods” need major revisions.

Taking all into in consideration, I recommend the publication of the manuscript in Nat. Comm., if the following concerns are addressed:

1. The equations punctuation need to be revised. For example, equation (2) ends with a dot and the next line starts with “where”.
2. Although there is no mistakes in equations (2) and (3), presenting both equations at the same time is confusing. Why not introduce each equation separately, explaining how each kind of anchoring can be achieved by tuning its relevant parameter? This would avoid, for example, the confusion made when homeotropic anchoring expression is showed, and later it is said “When the easy axis is along the surface normal, Equation 2 yields a homeotropic anchoring condition”.
3. In the second paragraph of page 4, the parameter ν is introduce without prior definition.
4. In page page 5, the authors define \tilde{Q}_{ij}^\perp as a projection onto the confining surface surface. I would suggest the authors to define this tensor as a projection to the plane perpendicular to ν (which would also justify the use of \perp in \tilde{Q}_{ij}^\perp). The projection onto the surface makes sense only if the we are dealing flat substrates, which is not the case. For spherical confining surfaces, the tensor \tilde{Q}_{ij}^\perp act as projection to the plane tangent to the interface.
5. In page 5 the authors used the same parameter n_e , without any further explanations, as the easy axis in the Rapini-Papoular model (Equation

- 2) and to define the angle θ_e in the conic degenerate free energy. While there is an easy axis n_e used to define the tensor Q^0 in Rapini-Papoular, the same concept can not be used for degenerate anchorings, where the free energy minimum lies in a path (or in a surface). I would advise the angle definition using other concept, or that the authors explain how n_e is defined in their free energy expression.
6. in Fig 1. A in page 6, the authors shows a schematic representation of the conical anchoring, however they do not show what the ellipsoids mean. Are the ellipsoids a representation of the liquid crystal director field in the surface? If so, I would not advise the use of a line of them pointing to the same direction, since the energy minimum degeneracy is an important feature of the authors model. Also, Figure b is missing a color scale.
 7. In equation 9, the authors defines an expression for the initial state of the liquid crystal director \mathbf{n} , but they do not mention with value is used for the order parameter S . If the order parameter used is S_{eq} , I suggest mentioning it in the text.
 8. In page 7, the authors mention the use of the Jones method to simulate the polarized light micrographs, but they do not mention how the optical setup is done. It would make the results easier to reproduce if the authors explain how the polarizer and analyser was set, and more important, which light wavelenght was used to simulate the micrographs.
 9. In page 9 the authors say “The vector field of forces in Fig 3b shows more clearly that the colloids attract each other for...”, however there is no mention in the paper how these forces are calculated.

Referee #1

Comment: **The Authors present an extension of their finding in ref 25. They apply Landau-de Gennes model to describe the director and scalar order at and around a sphere imposing a conically degenerate easy axis at its surface. Whereas ref 25 presented a discovery of a dodecapole particle as well as the main physical mechanisms and conditions behind its existence, the present paper cannot pretend for a similar level of impact and novelty. I therefore consider it as not suitable for Nature.**

Answer: *The Referee is correct in that the hexadecapole was experimentally discovered in Ref. 25. However, in this manuscript we go well beyond that initial observation. Specifically, we introduce a well-defined anchoring model that allows us to explain the existence of that hexadecapole at a quantitative level. Importantly, we also predict, for the first time, the existence of an elastic conic anchoring (CA) dipole, and its transition into a hexadecapole. Our predictions are confirmed by experiments. We have addressed the Referee's concern and emphasized the impact of this discovery in the introduction section (highlighted in blue).*

Comment: **Of course, this does not undermine the fact that the subject of the paper is very interesting. It presents a great piece of experimental work which has been performed at a very high level. However, I have found that the theoretical part and the general idea of conic degenerate anchoring as the necessary condition for the effect to occur should be revised. First, writing down an anchoring term does not account to a new model, the idea behind it is simple. In particular, calling \mathbf{P} a new projection tensor does not sound quite appropriate. This is a minor point.**

Answer: *We thank the Referee for the suggestion. We have changed the term used for \mathbf{P}' from 'a new projection tensor' to 'an operator tensor' in the first paragraph on Page 5 and the corrections are highlighted in blue.*

Comment: **The most serious problem is in the form of anchoring term (4). It is known that when all other forces are absent, the anchoring term is minimized by $Tet_s = Tet_e$, hence its expansion in $(\theta_s - \theta_e)$ must begin with a quadratic term. In contrast, the eq 7 produces a linear term $\sin(\theta_s - \theta_e)$ which is incorrect. The correct term, I believe, is $\sin^2(\theta_s - \theta_e)$.**

Answer: *We thank the Referee for raising this question, which provides a useful approach to validate our new proposed surface term. Assuming a small perturbation of the surface director tilt angle from the preferred*

tilt angle $\theta_s = \theta_e + \Delta\theta$, we have

$$\begin{aligned}\cos \theta_s &= \cos(\theta_e + \Delta\theta) \\ &= \cos \theta_e \cos \Delta\theta - \sin \theta_e \sin \Delta\theta \\ &\approx \cos \theta_e - \sin \theta_e \sin \Delta\theta\end{aligned}$$

And,

$$\begin{aligned}\cos^2 \theta_s - \cos^2 \theta_e &= (\cos \theta_e - \sin \theta_e \sin \Delta\theta)^2 - \cos^2 \theta_e \\ &= \sin^2 \theta_e \sin^2 \Delta\theta - 2 \cos \theta_e \sin \theta_e \sin \Delta\theta \\ &\approx -2 \cos \theta_e \sin \theta_e \sin \Delta\theta\end{aligned}$$

Thus,

$$\begin{aligned}f_{\text{surf}}^{c, \theta_s} &= \int_{\text{surf}} W_c S_{\text{eq}}^2 (\cos^2 \theta_s - \cos^2 \theta_e)^2 d\Sigma \\ &= \int_{\text{surf}} W_c S_{\text{eq}}^2 (-2 \cos \theta_e \sin \theta_e \sin \Delta\theta)^2 d\Sigma \\ &= \int_{\text{surf}} 4W_c S_{\text{eq}}^2 \cos^2 \theta_e \sin^2 \theta_e \sin^2 \Delta\theta d\Sigma\end{aligned}$$

As shown above, Equation 7 produces a quadratic term, $\sin^2 \Delta\theta$ or $\sin^2(\theta_s - \theta_e)$, which is the correct term, as suggested by the Referee.

Comment: Further, if \mathbf{S} can change then it can depend on the distance from the surface as well. Why strong deviations of n from n_e at the defect cores does not induce $\text{grad}\mathbf{S}$ along the surface normal to decrease the anchoring? To incorporate such $\text{grad}\mathbf{S}$ the anchoring term has to be modified into a bulk one.

Answers: Before we answer the above question, we would like to first clarify how the energy terms regulate the order parameter S and \mathbf{n} in the context of a Landau-de Gennes theory.

- Phase energy: assuming uniaxial ordering, the phase energy can be simplified into a function of S ,

$$f_{\text{phase}} = \frac{A}{3} \left(1 - \frac{U}{3}\right) S^2 - \frac{2AU}{27} S^3 + \frac{AU}{9} S^4.$$

The system is penalized when S deviates from the equilibrium order parameter S_{eq} , which is determined by parameter U .

- The elastic energy can be rewritten as

$$f_{\text{el}} = \frac{L}{2} \frac{\partial Q_{ij}}{\partial x_k} \frac{\partial Q_{ij}}{\partial x_k} = \frac{1}{3} L (\nabla S)^2 + LS^2 (\nabla \mathbf{n})^2.$$

It penalizes both ∇S and $\nabla \mathbf{n}$.

Figure 1: a) The evolution of scalar order parameter S near the particle at the equator plane along x axis.

- The surface energy reads as

$$\begin{aligned}
 f_{\text{surf}}^c &= \int_{\text{surf}} W_c \left(P'_{ik} \tilde{Q}_{kl} P'_{lj} - S_{\text{eq}} \cos^2 \theta_e P'_{ij} \right)^2 d\Sigma \\
 &= \int_{\text{surf}} W_c \left(S \cos^2 \theta_s - S_{\text{eq}} \cos^2 \theta_e - \frac{1}{3} (S - S_{\text{eq}}) \right)^2 d\Sigma.
 \end{aligned}$$

The surface energy is minimized when $S = S_{\text{eq}}$ and $\theta_s = \theta_e$.

The final configuration is governed by a balance between these three terms. Back to the Referee's question,

- ∇S does exist along the surface normal near the defect in our system, as shown in Fig 1.
- When a colloid with rigid anchoring is immersed in an LC host, the formation of a defect is inevitable. Near the defect core, where $\nabla \mathbf{n}$ starts to diverge, the system is forced to lower S in order to decrease the $LS^2(\nabla \mathbf{n})^2$ term in f_{el} , at the expense of f_{phase} and f_{surf} .
- In the extreme case of $S = 0$ (at the center of the defect core), the value of θ_s doesn't affect the surface term any more. In this sense, the effective anchoring decreases near the defect core. This effect results from the trade off between elastic energy and surface energy.

In conclusion, the above derivations serve to establish that our proposed model captures the physics near the defect very well, and already incorporates ∇S , obviating the need for additional new terms.

Comment: Finally, I see no reason for the special need of an azimuthally degenerate anchoring. This kind of anchoring is needed when the director turns about the surface normal to lower the energy cost. In the case of our interest this would mean to lower the cost of a fi director component. But in all of the figures both in the

present manuscript and in ref 25, I see the director to lie solely in the meridian planes without any fi component. Therefore, I conclude that to explain the effects presented it is sufficient to have a tilted easy axis lying in the meridional planes.

Answer: *We disagree with this comment: the degenerate conic anchoring model is essential to the results presented in our manuscript.*

- *Even though the surface directors lie solely in the meridian planes in our work, the surface director still has two different ϕ components, 0 and π , as shown in Fig 2. Only with these two ϕ components can one differentiate the hexadecapole and the elastic CA dipole configuration. As shown in Fig 3, the hexadecapole has different ϕ components between the two hemispheres, while the elastic CA dipole has only one. Moreover, the mechanism of the transition we observe is directly related to the “flip” of the surface director, thus the change from azimuthal angle ϕ between 0 and π .*
- *Due to the one-constant assumption adopted in our manuscript, the surface directors in our numerical calculations lie in the meridian plane, which is consistent with the report [Phys.Rev.E 84, 031702 (2011)] mentioned by the referee. The report also demonstrated the formation of a chiral dipolar component when the bend/twist anisotropy (k_{33}/k_{22}) increases. When k_{33}/k_{22} is set to 5, we observe a similar chiral CA dipole with $\theta_e = 45^\circ$, as shown in Fig 4. The angle ϕ is defined as the angle between the director and the meridian plane, which illustrates the twisting power or chirality near defects. Therefore, when taking into consideration the elastic anisotropy, the surface director may escape from the meridian plane, where a term which represents an anchoring degenerate in the azimuthal angle - as we proposed - is crucial in order to capture the resulting chirality. To address the Referee’s question, we have added the simulation result of chiral dipole with $k_{33}/k_{22} = 5$ and $\theta_e = 45^\circ$ to the supporting information.*
- *In addition, our new surface energy term can handle more complicated geometries, e.g. structures with no well-defined meridian plane.*
- *If the system has a fixed tilt easy axis on the meridian plane, as suggested by the Referee, the conic anchoring is then non-degenerate and the surface energy is minimized when $\theta_s = \theta_e$ and $\phi_s = \phi_e$. To model this mono-stable anchoring, the conventional uniform representation (Rapini-Papoular-like functional) should be used. As a consequence, the defect configuration will not respond to the reorientation of the far field in the same way as in Ref. 25 or our manuscript.*

To summarize, the new surface anchoring model proposed in our work is essential to produce the results presented in our work, and it provides a platform from which to explore more complex systems.

Figure 2: a) Schematic representation of θ_e and ϕ_s in degenerate conic anchoring

Figure 3: a-b) Sketch for director configurations on surface for colloids with degenerate conic anchoring ($\theta_e = 45^\circ$) for elastic hexadecapole (a) and elastic CA dipole (b).

Figure 4: a) Director fields of an elastic dipole with degenerate conic anchoring ($\theta_e = 45^\circ$) when $k_{33} = k_{11} = 5k_{22}$. The defect is shown in red. The angle ϕ (ϕ) is defined as the angle between the director and meridian plane.

Comment: Moreover, the anchoring can be just planar but not very strong that the cost of directors leaving the surface be compatible with the gain due to smoother distortions at the defect cores.

Answer: This is an interesting question; can a nematic colloid form a hexadecapole with a weak planar or homeotropic anchoring. As shown in Fig 2c of the manuscript, the n_x color maps for nematic colloids $\theta_e = 0^\circ$ and 90° both exhibit a quadrupole symmetry. A weaker anchoring strength will not change the symmetry, as shown in Fig 4 of Ref. 25. The formation of hexadecapolar symmetry arises from a superposition of the two quadrupoles of opposite sign, and is, therefore, not achievable by lowering the anchoring strength for either planar or homeotropic anchoring. Following the Referee's question, we have added this discussion in the supporting information.

Comment: In addition, I have not noticed specific means to get a conic anchoring at otherwise standard spherical particles. Is this so trivial? My knowledge has been that producing this kind of anchoring is a highly nontrivial problem.

Answer: We thank the Referee for this useful comment in relation to achieving conical anchoring on colloidal objects (this was done successfully over the course of 2 decades on flat surfaces). Recently, conical anchoring was shown to form spontaneously at the surface of polymeric spherical particles (ref 25), and colloidal magnetic nanoplates functionalized with polyethylene glycol with a well defined degree of polymerization and grafting density (Liu et al., PNAS 113, 10479, 2016) and isotropic glycerol droplets with a small amount (< 0.1 vol%) of sodium dodecyl sulfate, as described in the present manuscript. The method for obtaining spherical droplets with conical anchoring showing elastic hexadecapoles and conic anchoring dipoles in this present manuscript is rather simple, and sufficiently described in pages 12-13; this procedure can be easily reproduced following the description by multiple experimental groups. While there are other methods (for example, we have a separate manuscript in preparation, where conical anchoring is achieved by invoking electrostatic effects), we believe that describing these other approaches is beyond the scope of our present study. We have added a reference to [Liu et al in PNAS 113, 10479, 2016]. We trust that this, along with our response, fully addresses the questions of the Referee, for which we are grateful.

Comment: In this connection I may also note that an azimuthal fi director component (which would justify the degeneracy) on a sphere could give rise to a chiral dipolar component as described in Phys.Rev.E 84, 031702 (2011), and this would be experimentally detectable.

Answer: Thanks for the comment. As discussed above, consistent with the referred paper, the director stays in the meridian plane under the one-constant assumption in our manuscript. As the bend twist anisotropy (k_{33}/k_{22}) increases, we observe the chiral dipole configuration with $\theta_e = 45^\circ$ as well (Fig 4).

Comment: I also note that the fact that only three configurations around spherical particles with normal and planar anchoring are possible implies that the anchoring strength is infinite. If it is finite, then the configuration can be defectless. Or, maybe, of a hexadecapole type.

Answer: *As answered in the previous question, we cannot obtain a hexadecapole by merely weakening the anchoring.*

Referee #2

Comment: The authors of the manuscript develop a new elastic surface energy to model degenerate conic anchoring. Later, this expression is used to analyze the static liquid crystal director around a single colloidal particle, and after it, in a system composed of two of them. Finally, they study the transition between a state with dipolar defects to one with hexadecapolar one.

In general the article is well written. The abstract clearly shows the subject of the article and the introduction poses the problem very comprehensibly. The manuscript shows new results which are relevant to both basic and applied liquid crystal research. Moreover, all theoretical predictions are supported by experimental results. However, the section Model and Method need major revisions.

Answer: *We are grateful to the Referee's kind feedback. We have modified our manuscript with corrections highlighted in blue.*

Comment: The equations punctuation need to be revised. For example, equation (2) ends with a dot and the next line starts with where.

Answer: *Thanks for pointing out this mistake. We have corrected the equations punctuation in the manuscript.*

Comment: Although there is no mistakes in equations (2) and (3), presenting both equations at the same time is confusing. Why not introduce each equation separately, explaining how each kind of anchoring can be achieved by tuning its relevant parameter? This would avoid, for example, the confusion made when homeotropic anchoring expression is showed, and later it is said When the easy axis is along the surface normal, Equation 2 yields a homeotropic anchoring condition”.

Answer: *Thanks for the suggestion. We have separated the introduction of Equation 2 and 3. Also, we changed the order so that the surface normal ν is defined before its use in the homeotropic anchoring expression.*

Comment: In page page 5, the authors define \tilde{Q}_{ij}^\perp as a projection onto the confining surface surface. I would suggest the authors to define this tensor as a projection to the plane perpendicular to ν (which would also justify the use of \perp in \tilde{Q}_{ij}^\perp ?). The projection onto the surface makes sense only if the we are dealing at substrates, which is not the case. For spherical confining surfaces, the tensor \tilde{Q}_{ij}^\perp act as projection to the plane tangent to the interface.

Answer: *Thanks for the comment. We have changed ‘onto the surface’ to ‘onto the plane perpendicular to ν ’, as suggested by the Referee.*

Comment: In page 5 the authors used the same parameter \mathbf{n}_e , without any further explanations, as the easy axis in the Rapini-Papoular model (Equation 2) and to define the angle θ_e in the conic degenerate free energy. While there is an easy axis \mathbf{n}_e used to define the tensor \mathbf{Q}_0 in Rapini-Papoular, the same concept can not be used for degenerate anchorings, where the free energy minimum lies in a path (or in a surface). I would advise the angle definition using other concept, or that the authors explain how \mathbf{n}_e is defined in their free energy expression.

Answer: *Thanks for the comment. We agree that the use of \mathbf{n}_e can be confusing in the context of degenerate conic anchoring. We have modified the manuscript to avoid using \mathbf{n}_e for degenerate conic anchoring.*

Comment: in Fig 1. A in page 6, the authors shows a schematic representation of the conical anchoring, however they do not show what the ellipsoids mean. Are the ellipsoids a representation of the liquid crystal director field in the surface? If so, I would not advise the use of a line of them pointing to the same direction, since the energy minimum degeneracy is an important feature of the authors model. Also, Figure b is missing a color scale.

Answer: *Following the previous comment, we agree that the schematic representation in Fig 1a and use of \mathbf{n}_e can be confusing. Hence, we have replaced the image in Fig 1a to better represent the degenerate conic anchoring. Also, the color bar has been added in Fig 1b.*

Comment: In equation 9, the authors define an expression for the initial state of the liquid crystal director \mathbf{n} , but they do not mention with value is used for the order parameter S . If the order parameter used is S_{eq} , I suggest mentioning it in the text.

Answer: *Thanks for pointing out the missing information. We used S_{eq} to initialize \mathbf{Q} and we have added it to the manuscript.*

Comment: In page 7, the authors mention the use of the Jones method to simulate the polarized light micrographs, but they do not mention how the optical setup is done. It would make the results easier to reproduce if the authors explain how the polarizer and analyser was set, and more important, which light wavelength was used to simulate the micrographs.

Answer: *Thanks for the comments. We have added the description of the optical setup and light wavelength to the manuscript.*

Comment: In page 9 the authors say "The vector field of forces in Fig 3b shows more clearly that the colloids attract each other for...", however there is no mention in the paper how these forces are calculated.

Answer: *Thanks for pointing this out. The force is obtained by calculating the spatial gradient of the free energy. We have inserted it in the caption of Fig 3.*

Reviewer #1 (Remarks to the Author):

Second report of Referee #1

First of all I should apologize for the absolutely inappropriate comment on the validity of the anchoring term. Sorry, for some reason, which is now difficult to understand, I missed the fact that the difference of cosine squares in the integrand is in power two. I missed this two.

The authors have given a convincing reply on the necessity of resorting to a degenerate conic anchoring. I agree, in the case considered the anchoring has to be conic.

Overall, it is a good work which definitely deserves publication. As to specific venue of Nature, I used to think that this journal should avoid presenting further developments of previously published discoveries. The abstract says: In recent work, a new type of defect was discovered, the so-called elastic hexadecapole, which has been speculated to result from a conic anchoring condition at the colloid surface... In order to gain a systematic understanding of hexadecapoles and other related structures, a new continuum model for anchoring is introduced here at the level of a Landau-de Gennes free energy functional. Does not this mean a further development?

I might be naive in my expectations of publications in Nature. Therefore, I leave the decision on publication of this paper in Nature to the Editor.

Reviewer #2 (Remarks to the Author):

The author addressed all the concerns raised by me. I advise the publication of their work in Nature Communications.

Referee #1

Comment: **First of all I should apologize for the absolutely inappropriate comment on the validity of the anchoring term. Sorry, for some reason, which is now difficult to understand, I missed the fact that the difference of cosine squares in the integrand is in power two. I missed this two.**

Answer: *It was our pleasure to address the questions from the Referee. No need for apologies.*

Comment: **The authors have given a convincing reply on the necessity of resorting to a degenerate conic anchoring. I agree, in the case considered the anchoring has to be conic.**

Answer: *We are glad that the Referee concurs with the essential role of the degenerate conic anchoring condition in this scenario.*

Comment: **Overall, it is a good work which definitely deserves publication. As to specific venue of Nature, I used to think that this journal should avoid presenting further developments of previously published discoveries. The abstract says: In recent work, a new type of defect was discovered, the so-called elastic hexadecapole, which has been speculated to result from a conic anchoring condition at the colloid surface In order to gain a systematic understanding of hexadecapoles and other related structures, a new continuum model for anchoring is introduced here at the level of a Landau-de Gennes free energy functional. Does not this mean a further development? I might be naive in my expectations of publications in Nature. Therefore, I leave the decision on publication of this paper in Nature to the Editor.**

Answer: *We thank the Referee for his/her comments. As we have addressed in the manuscript, we have developed new terms in the context of a Landau-de Gennes free energy functional to capture the degenerate conic anchoring, and we have discovered new types of elastic dipoles and dipole-hexadecapole transformations. These results suggest the existence of previously unanticipated approaches to design self-assembled crystal lattice structure via the control of preferred tilt angle.*

Referee #2

Comment: **The author addressed all the concerns raised my me. I advise the publication of their work in Nature Communications.**

Answer: *We are grateful for the Referee's kind feedback.*